# Efficacy Study of Fault Trending Algorithm to Prevent Fault Occurrence on Automatic Trampoline Webbing Machine

**Shi Feng *** and **John P. T. Mo**

School of Engineering, RMIT University, Melbourne, VIC 3083, Australia; john.mo@rmit.edu.au
* Correspondence: fengshi19910310@gmail.com

**Abstract:** Nowadays, fault diagnostics is widely applied under Industry 4.0 to reduce machine maintenance costs, improve productivity, and increase machine availability. However, fault diagnostics are mostly post-mortem. When the fault is identified, it is already too late because damages have been done to the product and machine. This paper compares the efficacy of several signal data processing techniques for detecting faults that are about to occur. Our aim is to find an efficient way to predict the fault before it occurs. A continuous wavelet transform synchrosqueezed scalogram was found to be most suitable for this purpose, but it is difficult to apply. A novel procedure is proposed to count the number of pulses in the synchrosqueezed scalogram. A new method for detecting the trend from the pulse counts is then developed to predict the fault before it happens. The procedure and method are illustrated with experimental data collected while running an automated double-thread trampoline webbing machine.

**Keywords:** Statistical Process Control (SPC); Fast Fourier Transform (FFT); continuous wavelet transform (CWT); synchrosqueezed wavelet transform; scalogram; pulse time graphs





## 1. Introduction

Nowadays, the vast volume of sensor data from Industry 4.0 implementation enables fault diagnosis to play an important role in the manufacturing process as it has potential benefits to reduce machine maintenance cost, improve productivity, and increase machine availability. Many machine quantities can be captured in the Industry 4.0 environment as fault diagnosis parameters such as current, voltage, speed, vibration, and so on. Capturing these signals over time is the first step to determining if any process conditions have been changed [1]. Subsequently, digital signal processing methods are used to analyse different types of signals. Time-based signals can be processed by different methods which can be classified into three main categories: time domain, frequency domain, and hybrid time-frequency domain.

Time domain digital signal processing methods aim to detect abnormality in the signals over time. For example, using flight data as the foundation of discussion, Zhang and Zhang [2] reviewed basic time series analysis methods such as data filtering and trend forecasting. After appropriate processing, exception could then be detected according to the limits of the varying signal, and relevant actions could be taken to manage the problem.

Frequency domain methods analyse the transformed signal in the frequency domain. A common method is the Fourier Transform, which detects frequency components of the time-varying signal. Marcelo et al. [3] used the Fast Fourier Transform (FFT) technique for on-line failure detection of asynchronous motors. They detected and characterized frequencies in the motor's current signal as the fundamental rotational eccentric frequency, plus other frequencies that could be observed in the motor's operation. Detection of fault could then be asserted from the recognized patterns.

Hybrid time-frequency domain methods are primarily a mix of the time-based domain and frequency domain methods. Sepulveda and Sinha [4] developed a vibration condition

monitoring system using machine learning models on features derived from both time and frequency domain data. The primary goal was to imitate human experience and expertise in recognizing symptoms of fault. Hybrid time-frequency methods have the advantage of both time and frequency domain methods but suffer from the complexity and errors in simulated responses compared to experimental data.

Fault diagnostics research can be explored in two types: (1) identification of faults that have already occurred so that appropriate remedial actions can be taken to quickly rectify the problem [5]; (2) incipient fault detection to detect a machinery fault at its early stages and take actions before it develops into catastrophic faults during the manufacturing process [6]. Both types of fault diagnostics share similar analysis methods, but the approach and data requirements are different.

The first type of fault diagnostics methods aims to rectify problems. It is a post-mortem approach. The second type of incipient fault diagnostics is to detect if the machine, which is still working well, has the trend to develop a fault soon. It generally requires a good system model to support fault prediction. This is not always possible due to complexity or age of the system. On the other hand, using captured time-based signals, the time window for trend diagnostics varies greatly depending on the nature of the manufacturing process. Hence, the efficacy of analysis methods varies, affecting reliability of the diagnostics outcomes.

This paper uses an automatic trampoline webbing machine as the study platform to examine the efficacy of different methods. Signals captured by the digital sensors on the webbing machine were analysed, and fault trend indicated from the processed information was compared. Based on the evaluation, a new fault trending algorithm that combines digital techniques, including wavelet transform and scalogram, to determine if performance of the automatic webbing machine starts to deteriorate. The innovative system has the benefit of quickly classifying the fault signal type and identifying the time-frequency information so that the operator can manipulate the machine settings based on time-frequency information to avoid further faults happening.

## 2. Literature Review

The ability to detect the trend of the system to generate faults depends on finding a suitable method to assess the signal streams before a fault appears. Before examining a possible solution in the literature, four of the most-used signal processing methods in asset condition monitoring are reviewed.

### 2.1. Time Domain Analysis Methods

Time domain analysis is often affected seriously by noise in the signal. Pre-conditioning of the signals, especially high frequency signals, is an important task that can dictate correctness of the analysis outcome. Shim et al. [7] studied the hovering problem of an unmanned aerial vehicle. The time domain response data was processed in a linear time-invariant model of the vehicle. The analysis outcomes were used to improve stability of the flight control system. Likewise, Fu et al. [8] used an adaptive fuzzy clustering concept to group the vibration data into a data matrix with nine common time domain parameters. Exception processing could then be initiated according to the limits of the varying signal.

One of the important time domain methods is Statistical Process Control (SPC) [9]. SPC aims to monitor outcomes of a manufacturing process. If the process continues to produce good outcomes, no action is taken. However, if the process starts to produce rejects, a decision on whether to intervene will be taken. Mo et al. [10] used an SPC control chart to monitor the process deterioration caused by vibration in the manufacturing. An SPC control chart can also be applied to monitor the ecological system Shore [11]. This research demonstrated that SPC is a time-based out-of-limit detection system. Efficacy of the system depends on the limit value that needs to be set in such a way that it can raise the alarm when the system has the trend to produce more bad signals, but at the same time does not

raise the alarm too quickly while the system produces one or two bad signals occasionally (by chance). It is therefore a matter of risk of the system going bad too quickly.

### 2.2. Fast Fourier Transform (FFT)

Signals in the time domain are normally irregular and difficult to observe. Fourier transform solves the problem because such a transform converts the time domain information to the frequency domain and makes it more convenient to observe the irregular features. However, before the rapid development of computer technology, the application of Fourier was limited because the calculation was time-consuming. With the achievement of computer technology, FFT becomes one of the most widely used techniques in signal processing. Su and Lee [12] studied infrared spectroscopy signals during the scanning of the biological specimen to detect cancer. The frequency domain information then represented changing patterns in the tissue indicating the possible presence of cancer. FFT can be applied in the field of electrostatics; Ong et al. [13] implemented FFT to evaluate the multipole expansion.

Quality of the analysis outcome depends on the materials and components. Ding et al. [14] explored the application of FFT in the chatter detection during titanium alloys machining process, titanium alloys which are regarded as one of the most difficult metals to be machined because the material renders the tool blunt and broken. The research analysed the frequencies occurring during cutting. By eliminating the harmonics, more generic signals related to chatter were identified. However, the method is tedious and time consuming. Stanković et al. [15] reviewed the effect of instantaneous frequency in enhancing the signal-to-noise ration of the incoming signals. Fault conditions are then defined according to certain undesirable frequencies being detected.

Therefore, FFT is a broad spectrum analysis technique that does not easily point to specific symptom in the signal stream. To enhance its ability to determine more exactly when the fault occurs, FFT was applied with continuous wavelet transform to detect fault in the rotor–stator system [16]. This problem presents difficulties in fault trend diagnostics requirements, making it a feasible but rather tedious method to be used. Its advantage is simplicity of detection method, especially for signals that come from a sine-based oscillatory environment.

Fault diagnostics in single time or frequency domain has its limits as time and frequency information are both crucial in fault diagnostics.

### 2.3. Continuous Wavelet Transform (CWT)

Wavelet transform gives a good time-frequency representation in the processing of non-stationary signals. With the dilation and translation of the mother wavelet [17], wavelet transform can capture the time and frequency information of a given signal.

Wang et al. [18] reviewed the application of continuous wavelet transform (CWT) in the analysis of vibration data. They used the Gabor wavelet as the mother transform function. They demonstrated selection of parameters and supported new extensions of the wavelet transform method to overcome drawbacks while balancing time-frequency resolution trade-off. The Gabor wavelet was applied as a sliding window function to create short time Fourier transform (STFT) by the integration of the inner product of the sliding window function. The spectrogram, which was the squared absolute value of the STFT, could then be computed as indicators of faults in applications such as machinery monitoring [19] and acoustic localization of aircraft [20]. However, the fixed sliding window size has the possibility of losing high frequency information during signal analysis due to the fixed time and frequency information. Additionally, STFT is only suitable for stationary signal analysis, while most of the signal types in the manufacturing process are non-stationary. An alternative signal processing method is needed.

Teng et al. [21] developed a new demodulation analysis using Hilbert transform based on a time-frequency method known as empirical model decomposition. The new analysis method was applied to analyse the gearbox of a wind turbine with gear-pitting fault. They

were able to extract the fault modulation information without human intervention and with more exact information. Their work indicated that the readability of the CWT scalogram could be improved by the selection of different types of wavelets and noise interferences introduced in the system.

A CWT scalogram reflects the energy density of signals. Abrupt transition in signals result in a larger absolute value of the coefficient, which usually represents brighter colours in the scalograms. This is often used as the observation reference of the singular frequencies [22]. In the traditional CWT scalogram analysis, fault features are detected by CWT, the whole data set, and by observing the colour in the scalograms [23]. The readability of the CWT scalogram can be affected by the selection of different types of wavelets and noise interferences in the experiment. Li et al. [24] had a different opinion of the selection of waveforms in mechanical signal analysis. They justify their idea by comparing the performance of Haar CWT transform with Morlet CWT transform. They claim that Haar CWT transform has a better performance on the fault detection of gear crack and ball bearing. The result broadens the way of thinking about how to select the suitable wavelets, since the similarity between the wavelet and signal is not the only criterion. A CWT scalogram can be combined with a method such as empirical mode decomposition to predict respiratory disorders [25], detect faults in turbine gearbox [26], and monitor rolling bearing faults [27]. Furthermore, a CWT transform can be utilised as a feature extraction method to integrate with deep learning algorithms and build a system that can autonomously identify the fault at its incipient stage. Hasan et al. [28] applied CWT as a tool to convert the scalogram to grey-scale images and then apply an adaptive CNN (convolutional neutral network) to identify the pump condition.

Li et al. [29] developed a four-step method to identify weak fault information. Step 1 was to process multi-cycle signals with CWT. Step 2 was to obtain the corresponding scalogram, which was then analysed in Step 3 by reassigned wavelet transform. Finally, synchronous averaging was applied to the signals for every working cycle of the bearing in Step 4. The proposed method was found effective. This research demonstrated that the more advanced CWT scalogram processing has strong potential for trend detection due to its ability to highlight weak faults.

CWT plus scalogram presentation seems to be more suitable for fault trend analysis. However, its computation cost is high. A more powerful computational engine is normally required, which is most likely not feasible to be implemented on the tiny, embedded processor performing real-time control of the machine.

### 2.4. Reassigned and Synchrosqueezed Wavelet Transform

Josso et al. [30] introduced the theory and applications of a reassigned wavelet transform as advanced processing of a CWT scalogram. They discussed the drawbacks of the CWT and subsequent shortfall of analytical ability to distinguish abnormal signals clearly. The reassigned wavelet transform method originated from the work of Kodera et al. [31], who compared four methods of analysing time-varying signals. They found that the modified moving window method captured more energy content in the signal stream and was able to distinguish changes in time variations. Auger and Flandrin [32] further developed the reassignment method of wavelet transform, visualizing it as scalograms.

The objective of the reassignment is to change the geometrical centre of the window to the gravity centre so that the window can capture more energy content of the input signal. The principle of reassigned wavelet transform is to reassign the resolution window of the wavelet, which is the geometrical centre of the window, to be the gravity centre of the complex energy density distribution. Since a scalogram represents the average energy of the resolution window, the scalogram can only reflect poor energy density information at the geometrical centre. Application of the reassigned scalogram method makes the energy density contained in the reassigned scalogram more closely to the original inspected signal [33]. Furthermore, the better time and frequency concentration in the reassigned scalogram, the more the readability of the CWT scalograms improves [34].

Many algorithms have been introduced to improve the readability and the reliability of the conventional scalogram [23,35]. These algorithms are being pursued in order to increase the frequency and time resolution in the scalogram. Hence, reassigned wavelet transform has a better ability to distinguish characteristics of patterns in a signal stream.

Synchronsqueezed wavelet transform is a special case of reassigned wavelet transform but with added ability to re-construct the time domain data [36]. Daubechies et al. [37] described synchrosqueezed wavelet transform as a wavelet-based time-frequency realloca­tion method. It works by decomposing a signal into constituent components with time-varying oscillatory. Li et al. [38] combined reassigned scalogram with synchrosqueezed scalogram methods to find the incipient fault information of the rolling element bearing. Thakur et al. [39] showed that synchrosqueezing had a definite advantage in analysing noisy signals due to its robustness to bounded perturbations. Silva et al. [40] proved that a synchrosqueezed CWT scalogram is more robust in rub detection than discrete Fourier transform (DFT) and fast Fourier transform (FFT). Liu et al. [41] combined warping trans­form and synchrosqueezed transform to extract modal curves in shallow water waveguide. Synchronsqueezed transform was able to sharpen the frequency resolution, hence enabling more accurate characteristics to be identified. The method was applied to invert sea bot­tom parameters of a shallow water experiment. Mekaoui et al. [42] applied reassigned wavelet transform to mechanomyograms signals of patients. The reassignment wavelet transform improved resolution and readability of the plot of the scalogram. Subsequently, they were able to increase the sample size with a broader variety of muscular diseases. Mihalec et al. [43] reviewed synchrosqueezing procedure to identify damping of a vibrating system. They found that a frequency shift error existed due to numerical computation. They proposed three compensation strategies and demonstrated better localisation frequency localisation. Wang et al. [44] compared the effectiveness of synchrosqueezed wavelet with nonlinear squeezing time-frequency transform in the rotor rub-impact fault detection. Af­ter analysing different time-frequency analysis (TFA) algorithms, Yu et al. [45] proposed a method named second order multi-synchrosqueezing transform, which improved the energy concentration of the traditional synchrosqueezing transform, thus revealing more critical information in rub-impact detection.

Due to better localization effect, synchronsqueezed wavelet transform, being a spe­cial kind of reassigned wavelet transform, provides a promising base for a method that can enrich the signals locally, allowing easier quantification of eventual signal trend to be highlighted.

## 3. Experimental Settings

Experiments were conducted on an automatic trampoline weaving machine. Similar to the traditional loom, two manufacturing processes take place on the machine: the warp preparation and weft insertion. Our experiment focused on the faults that occurred during the weft insertion process. There are two sets of hooks at both sides of the machine. Figure 1 shows the hooks mounted on the strainer. The thread is tied to the other side in similar fashion. This process is currently manual. Figure 1a,c display the strainer and hooks on the machine, while Figure 1b,d show the rack and pinion mechanism, and how the yarn threads are tied to the hooks.

The next step is the weft insertion process, which is equivalent to weaving. When the weft is picked from the weft package, the rapier will carry the weft to cross the width of the warp. The thread is hooked at the end of the rapier movement so that a double threaded weft is formed. The comb pushes forward to lock the double to the trampoline. The two heddles change its position ready for the next weft.

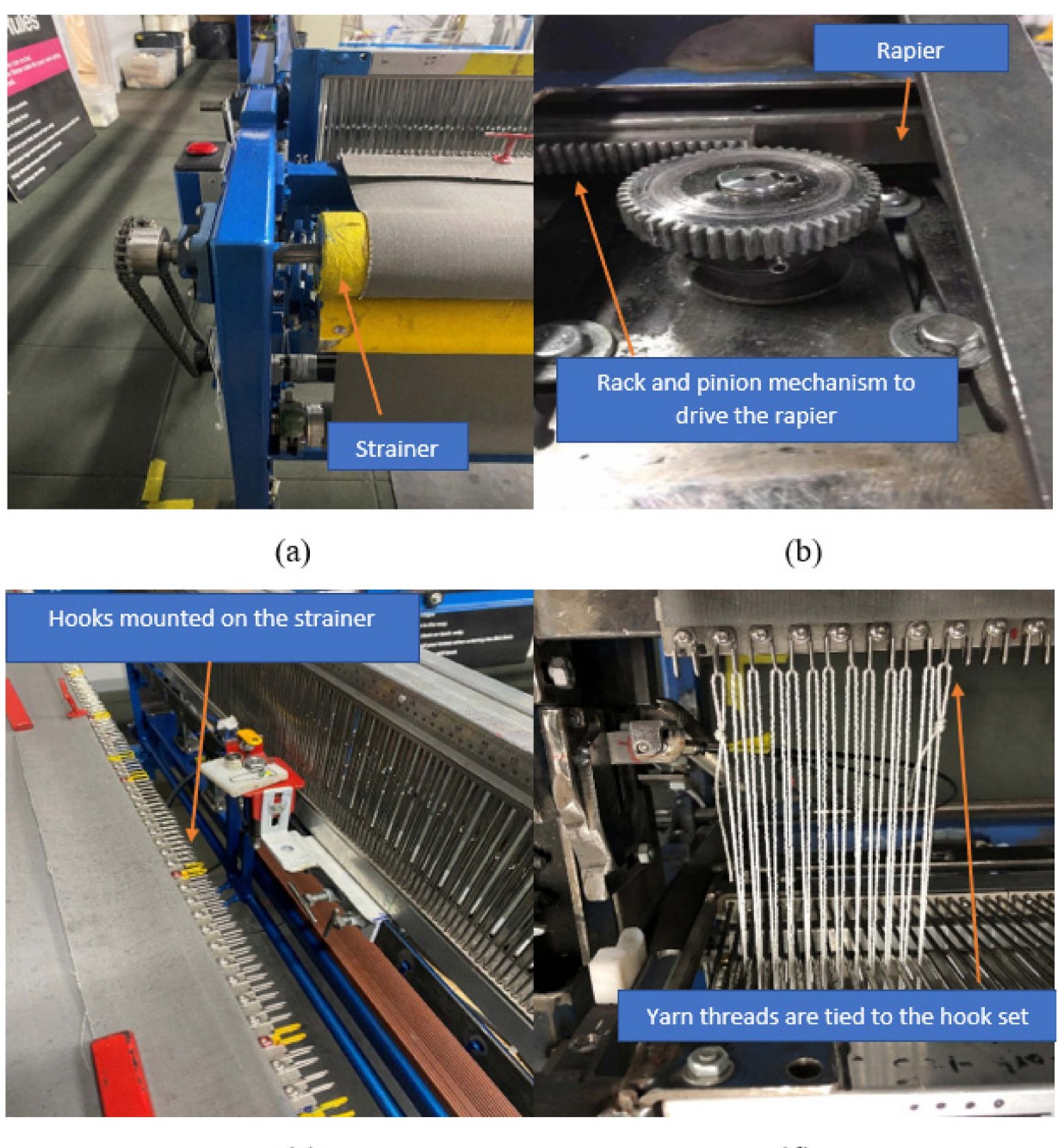

**Figure 1.** Hooks are mounted on a strainer: (**a**) The strainer rotates to wind the trampoline mattress during the weaving process, (**b**) rapier is driven by a rack and pinion mechanism, (**c**) hooks are mounted on a strainer, (**d**) yarn thread is tied to the hook one by one at both sides of the machine.

The weft insertion process has been automated [46] such that the rapier is controlled by a stepper motor; the rapier then repetitively carries the weft yarn from one side of the machine to the other, completing the weft insertion process. Although the stepper motor and the rack mechanism provide a strong support for the rapier's movement, the slip and dislocation between rack and rapier as well as the missing hook on the other side of the machine still occur randomly. In a manual machine, this missing hook will be observed by the operator and corrected immediately. However, in the new automatic machine, if the missing hook occurs in the middle of the matt, the machine will continue. Unfortunately, the correction of the missing hook requires that the matt be de-threaded until the missing

hook point and re-manufactured from that point. The re-work process is extremely time consuming and distressed.

The objective of this paper is to find a trend detection method that can predict the fault during weft insertion process. This trend detection method should indicate that the system is on the path of approaching a fault condition so that the operator knows the exact time and can take appropriate action. To achieve this, a signal collecting system is needed on the machine, in this case, a vibration signal on the machine's major moving part, i.e., the rapier.

To simulate the onset of a fault, the guide to the yarn was slowly tightened. This situation is a common scenario when undesirable friction is built up over a period inducing different tensional forces. The forces were very small, but the movement of the rapier was slightly affected. Hence, it is expected that analysing vibration signals on the rapier before and after a fault occurring could provide the basis for comparing efficacy of different signal diagnostic methods detecting the trend of faults.

To acquire signals from the manufacturing process, an LSM 303DLHC accelerometer on an Arduino Mega 2560 board is installed on the rapier (Figure 2). As the rapier moves back and forth, vibration signals are transmitted to the host station by Wi-Fi. Twenty minutes was the time length of one data set. Each data set therefore contains more than 2000 points. The experiments were repeated to acquire a reasonable number of data sets. A data record was marked normal if there was no fault observed. A data record was marked abnormal if faults such as a missed hook or loose thread was observed. These data sets were then processed by the algorithms described in the following sections.

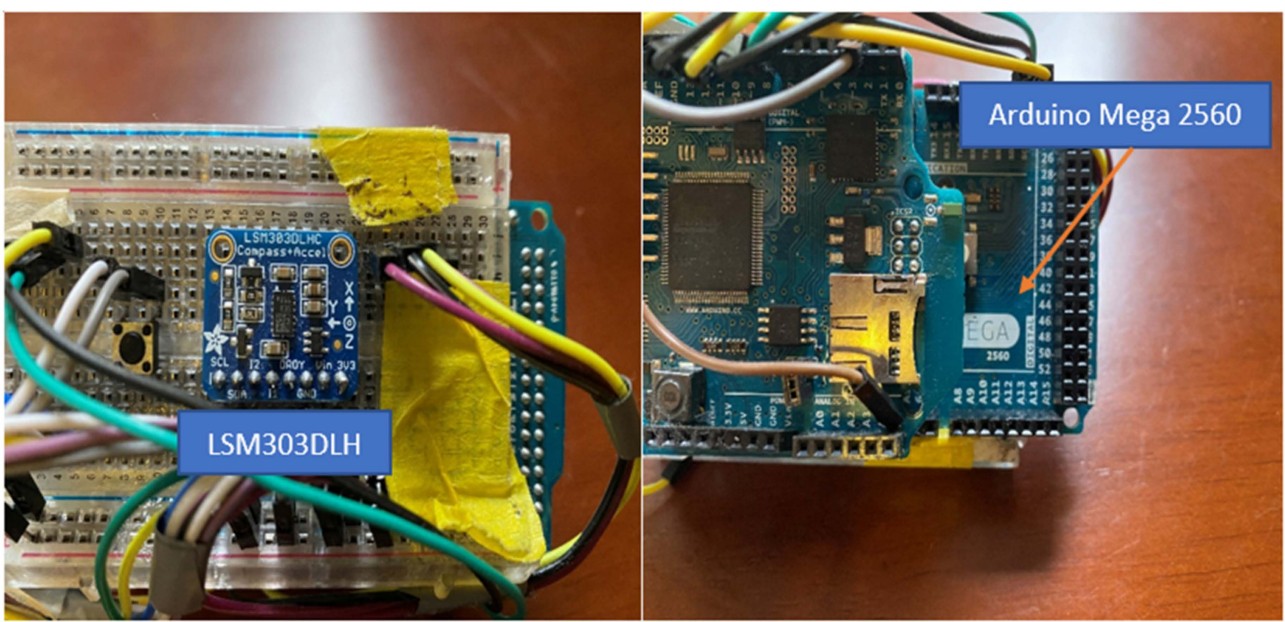

**Figure 2.** Experimental setting—LSM 303DLHC accelerometer.

## 4. Analysis Using Time Domain Method

To analyse the data sets in the time domain, the simplest time domain method, i.e., SPC control chart was used. Figure 3a shows the control chart for vibration on one of the directions of a normal data type. The horizontal axis in Figure 3a,b represents the time when the data points are collected. The vertical axis of the upper part is the sample mean of the data points, and the lower part is the standard deviation of the data points. There are two red lines in the Figure 3, which are the upper and lower control limits. Normal data points should be located within these two lines. If two consecutive points are out of the control limits, there is a fault-occurring trend. However, we can observe that many red points are out of the control limits, especially in the standard deviation of the data points in Figure 3a. Figure 3b is an abnormal signal pattern; during the data collection,

no fault occurred at the beginning of this abnormal data set. However, a lot of red points are out of the control limits in Figure 3b when the data collection process starts. Since out-of-control points are also common in "normal" operations, the fault trend is not clearly distinguishable from the acquired data sets. Therefore, it is concluded that the control chart is not an efficacious algorithm for this research.

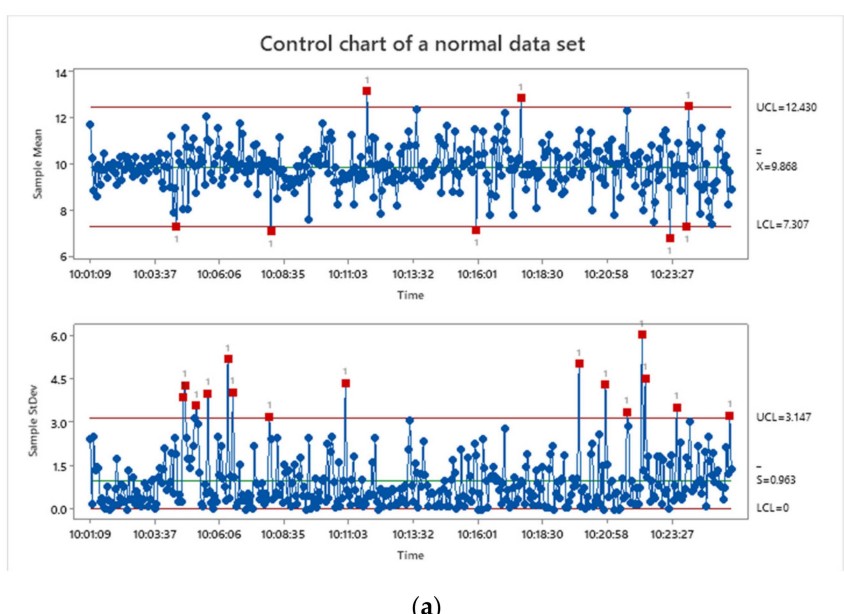

(**a**)

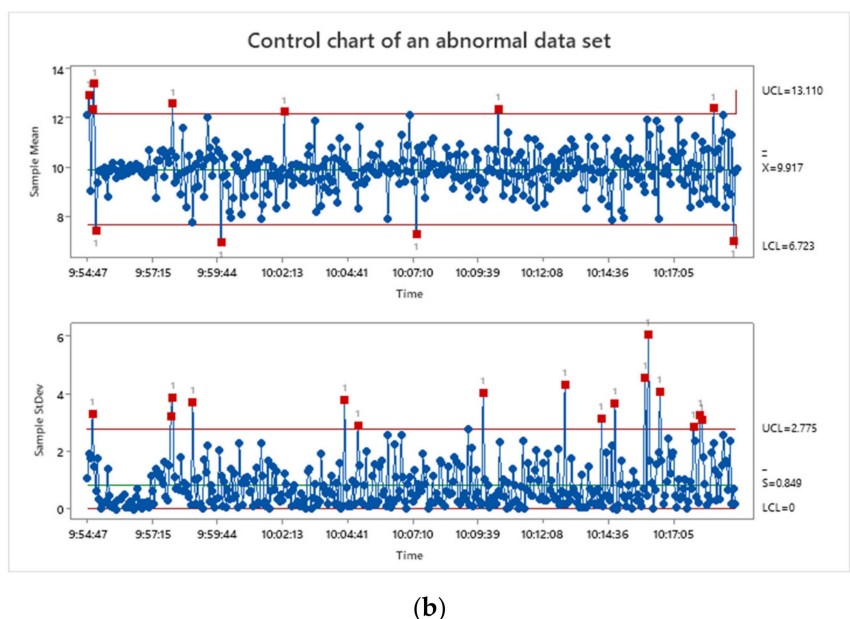

(**b**)

**Figure 3.** SPC control charts of two different data sets: (**a**) control Chart of a normal signal pattern, (**b**) control Chart of an abnormal signal pattern.

## 5. Analysis Using Fast Fourier Transform (FFT)

Numerous datasets were analysed using FFT functions on MATLAB. FFT transformed the acceleration data into frequency domain. If a fault occurred, some strange frequency points could be observed.

Figures 4 and 5 are generated by the whole set of the vibration data of FFT. Figure 4 is a set of normal vibration data while Figure 5 is an abnormal data set. It can be seen that both figures do not have uniquely identifiable frequencies to distinguish them from the other data set.

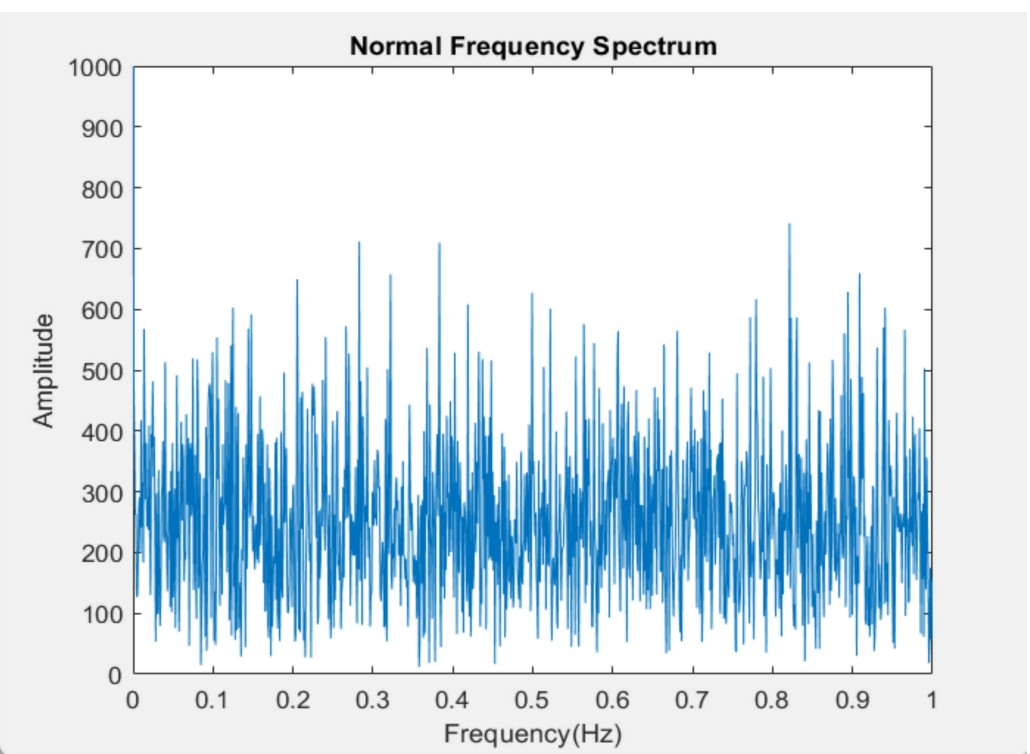

**Figure 4.** FFT data of a "normal" data set.

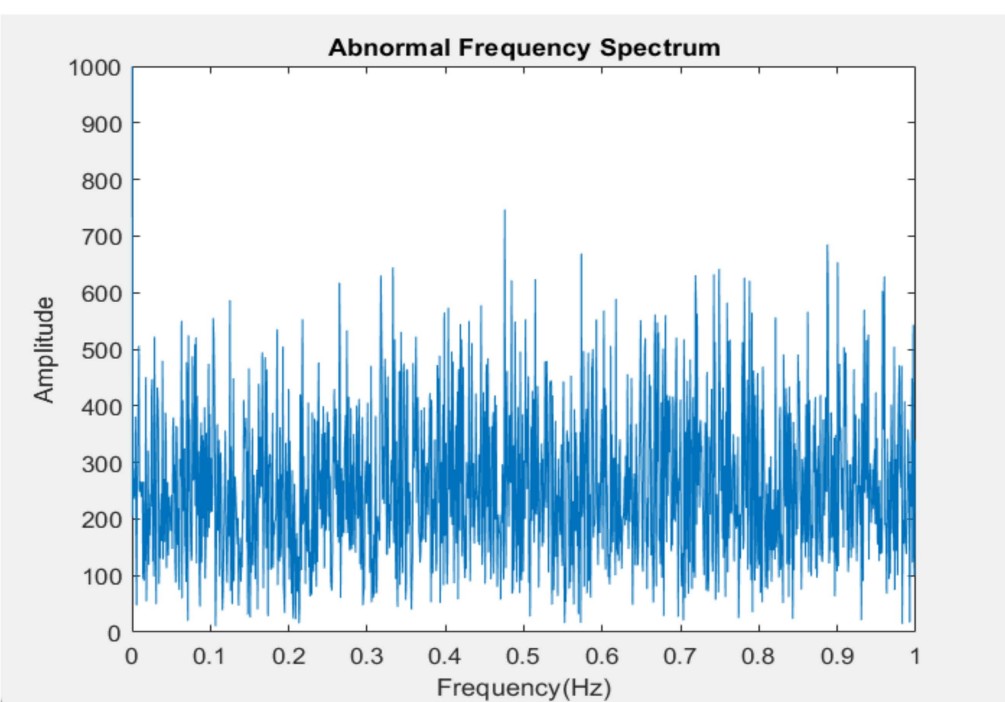

**Figure 5.** FFT data of an "abnormal" data set.

Although there is fault information contained in Figure 5, it is difficult to observe the singularity. The fault in this case is not recognisable under FFT processing. There is no feature on FFT that can distinguish the change from "normal" to "abnormal" conditions. This means that the efficacy of FFT to detect the trend of process moving to faulty situation is insufficient.

## 6. A CWT Fault Prediction System

Many researchers have applied a CWT scalogram to identify the fault features in signals, and some of them have achieved promising results. However, there still exist problems when dealing with transient changes in the signals, especially the whole data set of CWT. Useful information may disappear because of energy leakage or poor frequency resolution happened during CWT of the whole data set.

The proposed data processing method cuts the whole data set into segments. By choosing the right scale for small segments, transient and unnoticeable information can be extracted when CWT is applied to the whole data set over a number of time segments. Since each data set contains 2000 more points as mentioned (about 20 min), the question is: how many smaller time segments within these 20 min should be used to detect the changing machine conditions. After numerous trials of dividing the data set in different ways, it was found that the most effective length was around 300 to 350 points when the frequency information could be explicitly observed on the scalograms. To create continuity as the detection process moves between time segments, 50 data points are duplicated between every 350 points, resulting in a total of more than 2400 points. Hence, each data set was divided into seven sub-time segments, that is 1–351 to 2101–2451. The sampling frequency was two points per second. Thirty-one scalogram images were generated for each 350-point length segment.

Figure 6 shows four of the 31 scalograms in one sub-time segment. The absolute coefficient value represents the colour in the scalogram changing from below 0.5 to above 3.5. The values are indicated by the colour scale bar at the right-hand side of Figure 6. Each high value region (yellow regions) represents one single pulse.

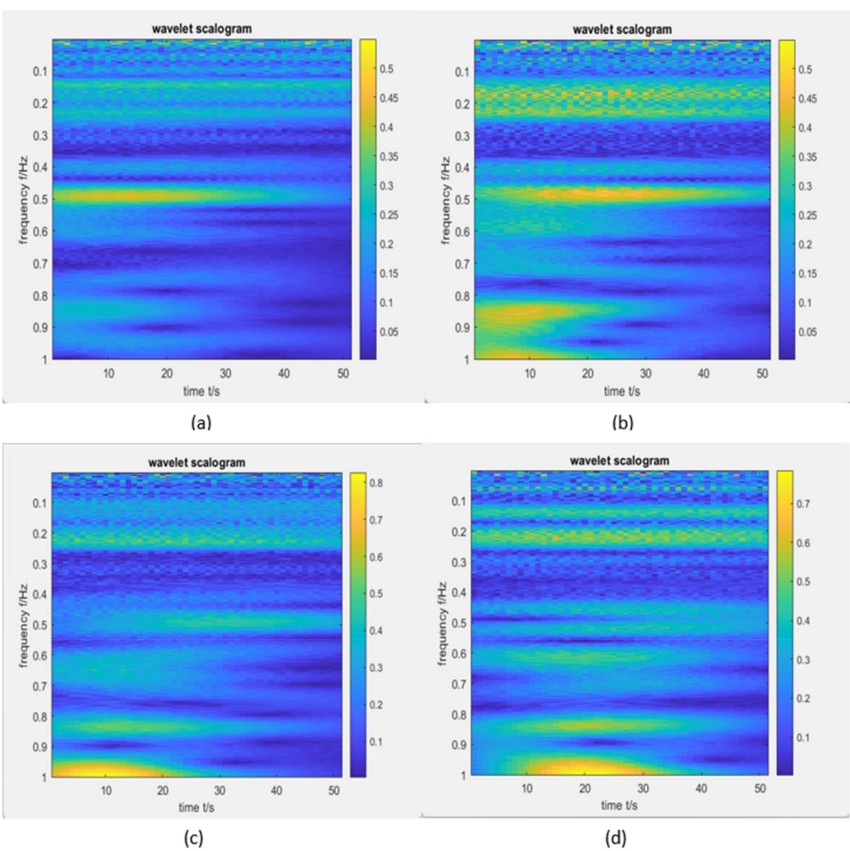

**Figure 6.** Four of the 31 CWT scalograms in one semi-time segment: (**a**) fuzzy pulses in frequency ranges 0.4–0.5 Hz and others, (**b**) fuzzy pulses in frequency ranges 0.4–0.5 Hz, 0.8–0.9 Hz, 0.9–1.0 Hz and others, (**c**) fuzzy pulses in frequency ranges 0.9–1.0 Hz and others, (**d**) fuzzy pulses in frequency ranges 0.8–0.9 Hz, 0.9–1.0 Hz and others.

The scalograms in Figure 6 are typical CWT scalograms obtained from the trampoline machine. When a yellow colour bar appears, it means that the shape of the wavelet matches the shape of the signal in this time segment, thus creating a larger value of coefficient and brighter colours. However, the scalogram pattern changes over the 20 min operation duration. Therefore, it is still necessary to find how the pulses in the frequency range trigger the faults that occur in the real world by analysing further the information presented in the scalograms.

It is hypothesized that there was a relationship between the number of pulses and the frequency ranges that these pulses were found to have. Changes in the number of pulses over time for these frequency ranges could provide an indication of the trend to onset of a fault. However, counting pulses in the scalogram depicted in Figure 6 is difficult because the high coefficient values are not focused. A more precise distinguished frequency indicator at different times is required to provide a clearer recognition of fault trend. In this case, a synchrosqueezed scalogram is introduced to further improve the readability of the scalogram and the efficiency of the pulse counting process in different special frequency ranges.

## 7. Synchrosqueezed CWT Scalogram Fault Prediction System

Synchrosqueezed wavelet transform has a better ability to distinguish characteristics of patterns in a signal stream. It is therefore logical to develop a more structured approach to use synchrosqueezed wavelet transform to extract "weak" faults before these faults can explode into real faults.

### 7.1. Proposed Fault Prediction System

The proposed method uses CWT functions in MATLAB and consists of four steps:

- First, the time domain signal streams are read into the system to generate wavelet scalograms. Scalograms are graphs that can reflect the similarity between wavelets and analysed signals and thus find the singular frequencies in the signals.
- Second, the centre of gravity of the complex energy density distribution is adjusted to maximise the energy density of within the resolution window. The data can then be exported to a data sheet, such as Excel, for analysis.
- Third, the pulse distribution in the synchrosqueezed wavelet transform data is counted in suitable groups, such as a histogram. The number of pulses in each frequency range is an indication of the pattern of operation at the time window. To assist the visualisation of the process, a radar graph with a different time window is plotted. It is necessary to note that plotting the spider graph is purely a visualisation exercise.
- Lastly, the number of pulses in the dominant frequency ranges is checked against time. Characteristics of the changes in the number of pulses in the dominant frequency ranges are observed and associated with a fault recorded for that data set.

### 7.2. Synchrosqueezed Wavelet Transform

The aim of the synchrosqueezed transform is to retrieve the time-varying amplitudes and the instantaneous frequencies in the signal. Therefore, synchrosqueezed transform is applied to improve the readability of the scalogram due to the elimination of the interference factors.

In Figure 7, the same colour scale bar applies. The locations for high value pulses are significantly concentrated. The readability for pulse counting is improved as the pulse in the scalogram is easier to identify.

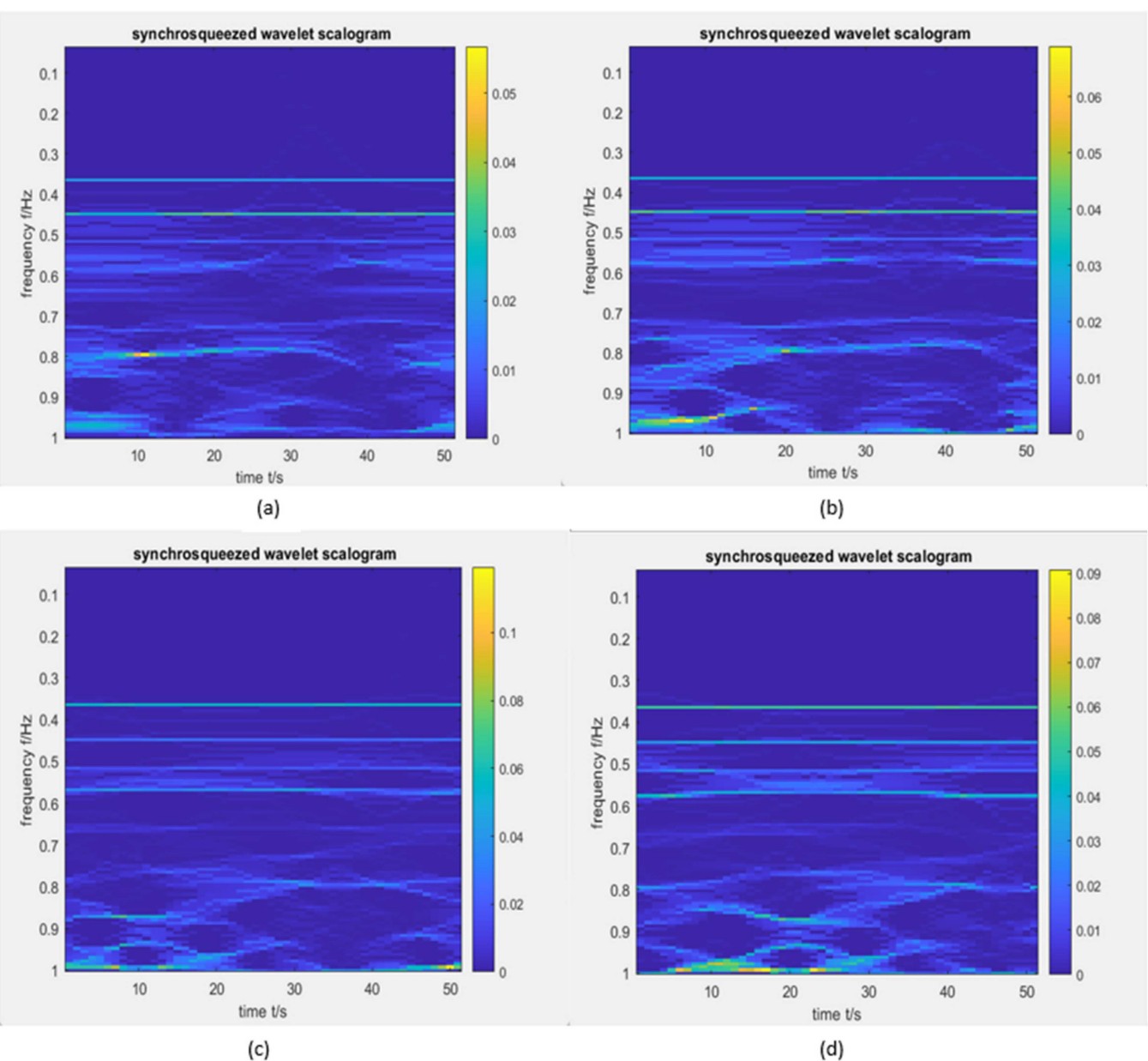

**Figure 7.** The synchrosqueezed wavelet transform synchrosqueezed scalograms of the four CWT scalograms shown in Figure 6: (**a**) more concentrated pulse in frequency range 0.4–0.5 Hz, and a new pulse was found around frequency 0.8 Hz, (**b**) more concentrated pulses in various frequency ranges, (**c**) more concentrated pulse in frequency range 0.9–1.0 Hz, (**d**) more concentrated pulses in frequency ranges 0.8–0.9 Hz and 0.9–1.0 Hz, moreover, the time information is explicit compared with Figure 6d.

*7.3. Analysis of Synchronsqueezed Data*

The synchrosqueezed scalograms were counted in each time segment. After counting the number of pulses in different frequency ranges, the frequency range distribution can be drawn. Figure 8a shows the number of pulses in different frequency ranges of each time segment in an abnormal data set. Figure 8b is a normal data set of frequency range distributions in each time segment.

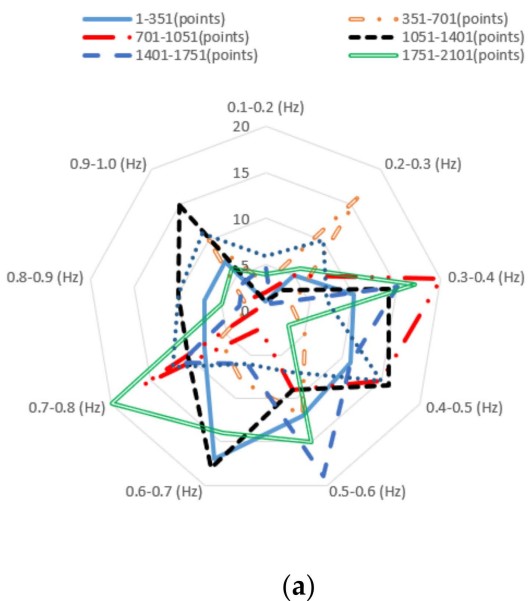

(**a**)

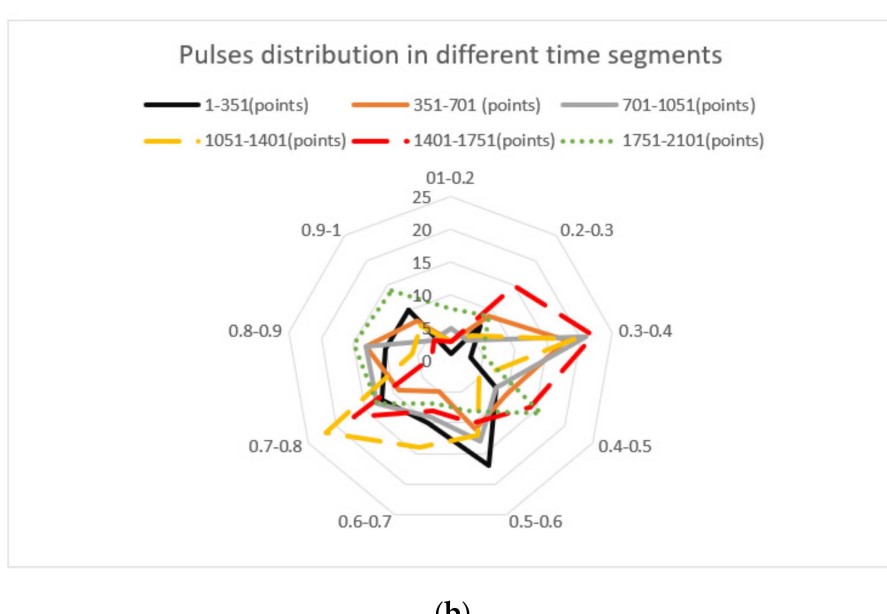

(**b**)

**Figure 8.** Pulse distribution in different time segments of two data sets: (**a**) pulse distribution of frequency ranges in different time segments (Abnormal data set), (**b**) pulse distribution of frequency ranges in different time segments (Normal data set).

There must be some singular pulse distributions in a data set that contains fault information. One hundred more data sets were acquired over the space of a week. Approximately half of these data sets were "normal" (i.e., no fault) and the other half were "abnormal" (i.e., with some faults induced somewhere in the duration). After analysing these data sets, it was found that the number of pulses in frequency range 0.3–0.4 Hz and 0.7–0.8 Hz were highly simultaneous in both types of data sets. This phenomenon can be observed from the pulse distribution graphs in Figures 9 and 10. There are numerous time segments containing the above-mentioned feature, and we selected eight similar features in different time segments of six data sets (three normal and three abnormal), then demonstrated in Figures 9 and 10.

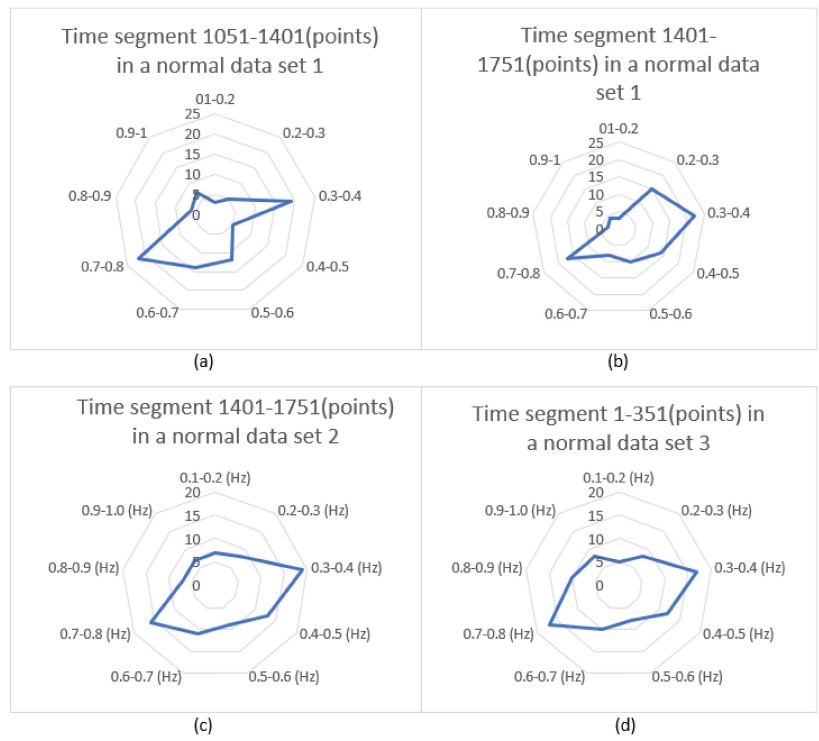

**Figure 9.** Number of pulses in different frequency ranges for "normal" data sets: (**a**) feature 1 in time segment 1051–1401 (points) of data set 1, (**b**) feature 2 in time segment 1401–1751 (points) of data set 1, (**c**) feature 3 in time segment 1401–1751 (points) of data set 2, (**d**) feature 4 in time segment 1–351 (points) of data set 3.

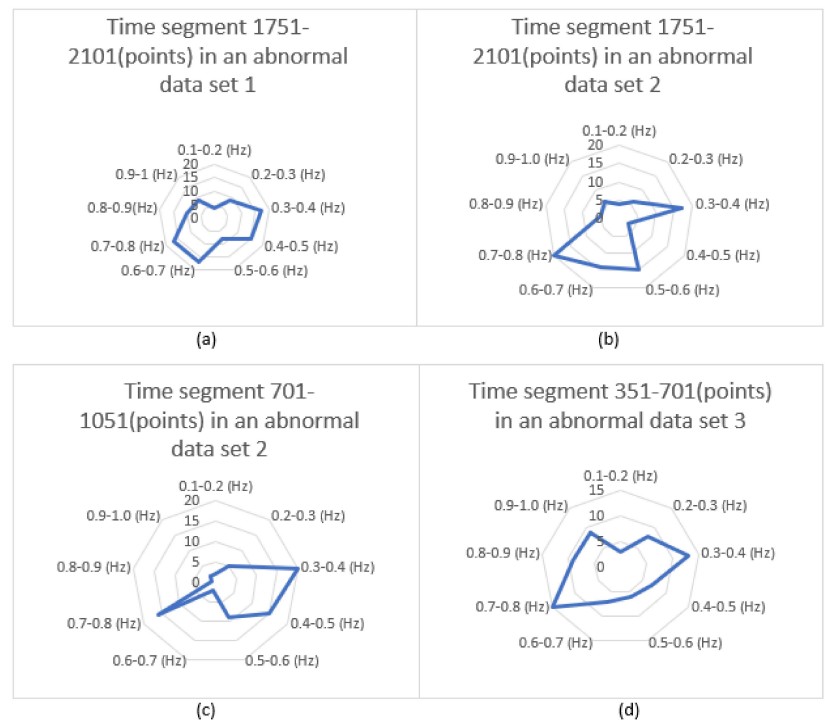

**Figure 10.** Another set of number of pulses in different frequency ranges for "abnormal" data sets: (**a**) feature 5 in time segment 1751–2101 (points) of abnormal data set 1, (**b**) feature 6 in time segment 1751–2101 (points) of abnormal data set 2, (**c**) feature 7 in time segment 701–1051 (points) of abnormal data set 2, (**d**) feature 8 in time segment 351–701 (points) of abnormal data set 3.

All of the data sets obtained experimentally exhibited similar pattern, including those with fault recorded, with the number of pulses in 0.3–0.4 Hz and 0.7–0.8 Hz reaching a large number simultaneously in both data types. Compared with other frequency ranges, the two frequency ranges were found to be different from the others, because the distribution of pulses in these ranges has the previously mentioned regularity, and it has repetitively appeared in both normal and abnormal data sets. For easier explanation, these two frequency ranges were named as "dominant frequency ranges".

### 7.4. Fault Prediction

In order to observe the fluctuation of the two dominant frequency ranges change with time, the number of pulses in the frequency ranges 0.3–0.4 Hz and 0.7–0.8 Hz is plotted over time segments, as shown in Figure 11a,b.

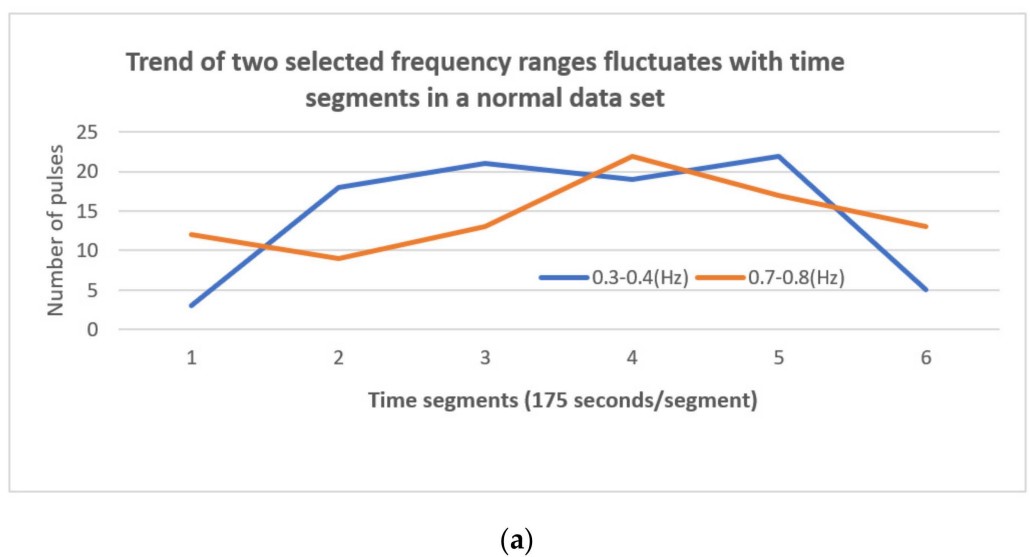

(**a**)

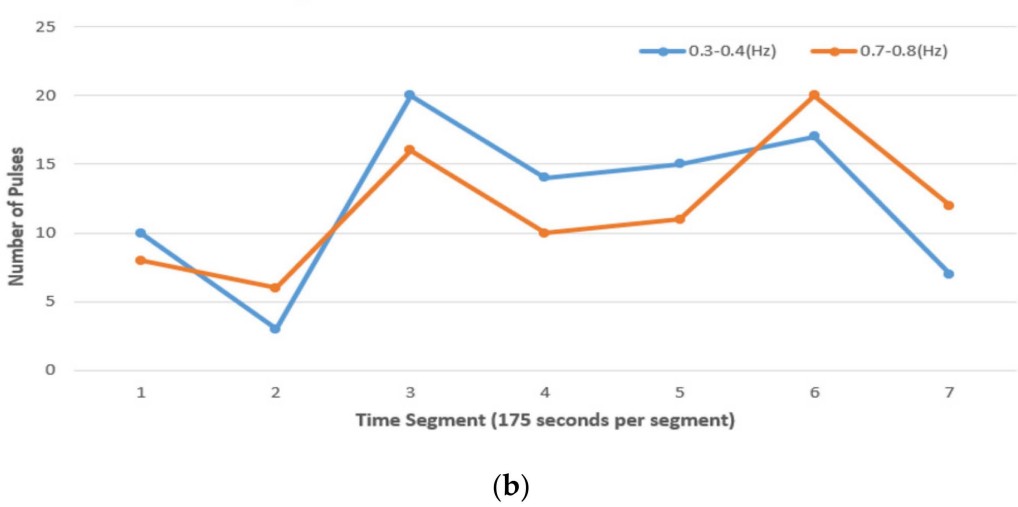

(**b**)

**Figure 11.** Trend of two selected frequency ranges fluctuates with time segments in two data sets: (**a**) number of pulses in the two dominant frequency ranges along time segments in a normal data set, (**b**) number of pulses in the two dominant frequency ranges along time segments in an abnormal data set.

In Figure 11a,b, the number on the horizontal axis represents the time segment. These time segments, combined with the points on the line graph, can be compared with the recorded fault-approaching time. By comparing the pulse change of 0.3–0.4 Hz and 0.7–0.8 Hz in both normal and abnormal data sets, a phenomenon has been noticed when the fault is approaching, that is, the number of pulses in the two selected frequency ranges simultaneously reach a turning point in the fault-occurring time segment. To be more precise, the number of pulses in both frequency ranges will drop first to a turning point, then start to increase. This trend can be observed in time segment 2 and 4 of an abnormal data set. In normal data sets such as Figure 11a, such a phenomenon did not occur.

To further confirm the phenomenon, other sets of normal and abnormal data have been collected to generate radar charts and line graphs. Figures 12 and 13 have been chosen as examples of demonstrating the result:

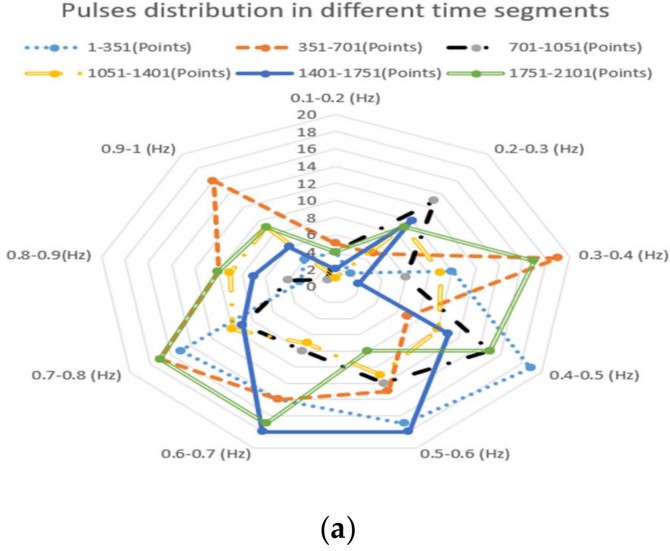

(**a**)

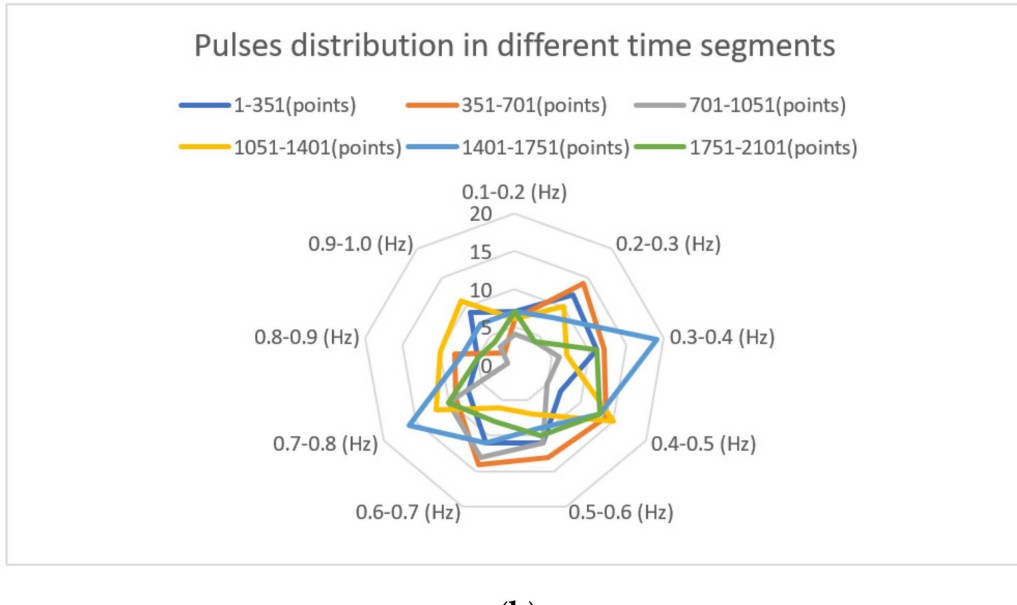

(**b**)

**Figure 12.** Pulse distribution in different time segments of another two data sets: (**a**) pulse distribution of frequency ranges in different time segments of an abnormal data set, (**b**) pulse distribution of frequency ranges in different time segments of a normal data set.

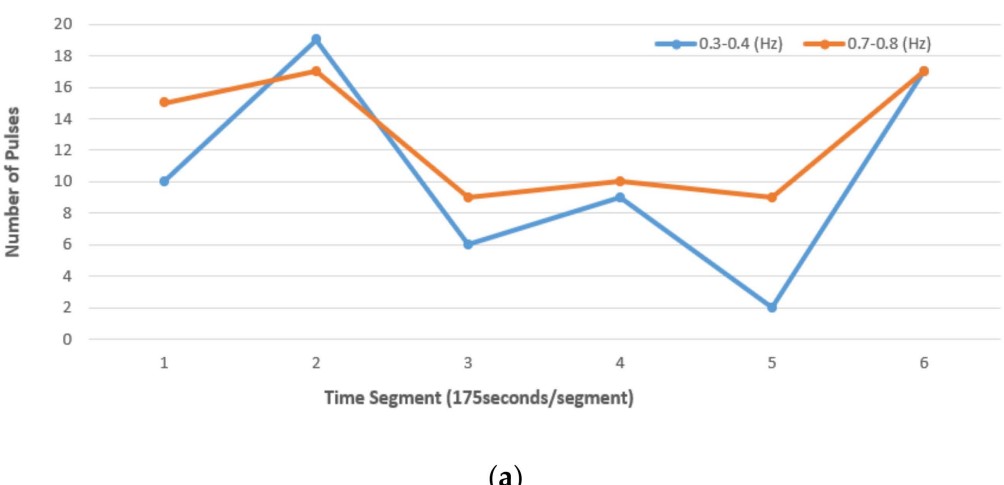

(**a**)

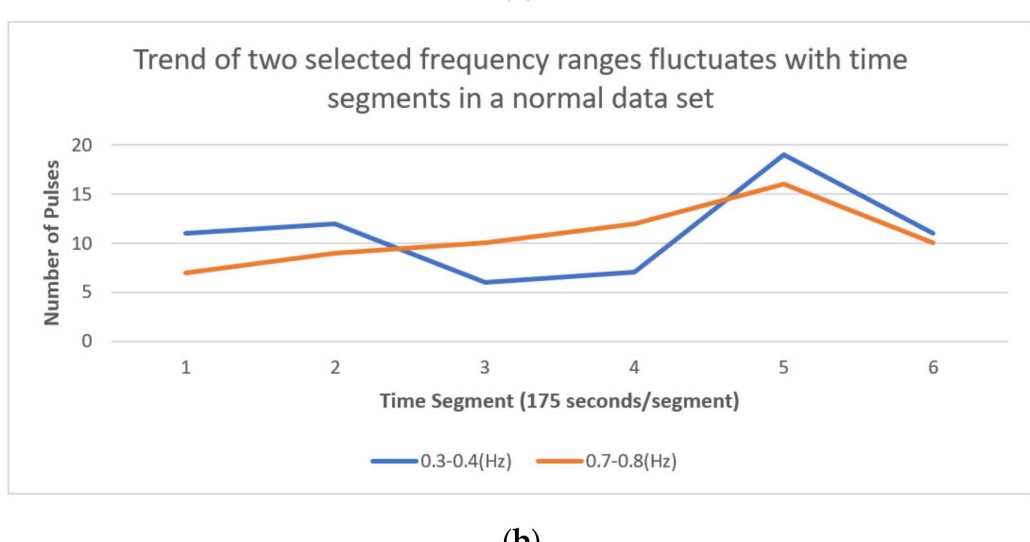

(**b**)

**Figure 13.** Trend of two selected frequency ranges fluctuates with time segments in another two data sets: (**a**) number of pulses in the two dominant frequency ranges along time segments in an abnormal date set, (**b**) number of pulses in the two dominant frequency ranges along time segments in a normal data set.

Again, the dominant frequency ranges of 0.3–0.4 Hz and 0.7–0.8 Hz were observed. The number of pulses in these frequency ranges were plotted against time segments as shown in Figure 13a,b.

In the abnormal data set, such as Figure 13a, the decreasing and increasing trend fluctuation happened around time segments, i.e., time segment 3 and time segment 5; however, in normal data sets, such as Figure 13b, no such trend fluctuation can be found. Hence, through the trials of numerous data sets, the dominant frequency ranges are identified, the number of pulses can be clearly extracted from synchrosqueezed wavelet transform, and the trend of fault conditions can be detected in the time segments when the fault is upcoming.

## 8. Discussion

The analysis process described in Sections 6 and 7 shows that the CWT scalogram algorithm identifies the fault by simply transforming (by CWT process) the whole abnormal data set and then comparing the result with the scalogram of the entire normal data set. The method is applicable with preconditions. The first precondition is that the fault must have already happened; a CWT scalogram algorithm cannot detect the fault signal if the fault does not exist, i.e., the data set recorded stops before a fault occurs. The second precondition is that the occurred fault must be visible from the data set after CWT transform. However, the two preconditions are not established in the trampoline machine situation. The pulse change is weak and basically not detectable in the range of CWT frequencies. Besides, our goal is to predict the fault before it happens. Therefore, the proposed method in this research zooms into the signal by cutting the data into sequential time segments. However, the length of time segments affects "resolution" of detecting a fault somewhere in the series of scalograms. Therefore, CWT scalograms and synchrosqueezed scalograms in Figures 6 and 7 are scalograms that contain pulse information of the machine at different time segments. These diagrams still require further processing according to their respective time segments before the data set can be testified as normal or abnormal. This further processing is achieved by observing changing number of pulses as described in this paper.

Having introduced the proposed data processing method, we also want to validate the result of segmented FFT. We choose the same segmented time interval shown in Figures 6 and 7, i.e., 350 data points with 50 data points overlap. We generate numbers of segmented FFT figures. Figure 14 shows 12 spectrums generated by segmenting the data sets that produced Figures 4 and 5, then we select 6 consecutive segmented FFT spectrums, respectively, from Figures 4, 5 and 14a,f are segmented normal FFT spectrums, while Figure 14g,l are abnormal segmented FFT spectrums. The results are demonstrated below.

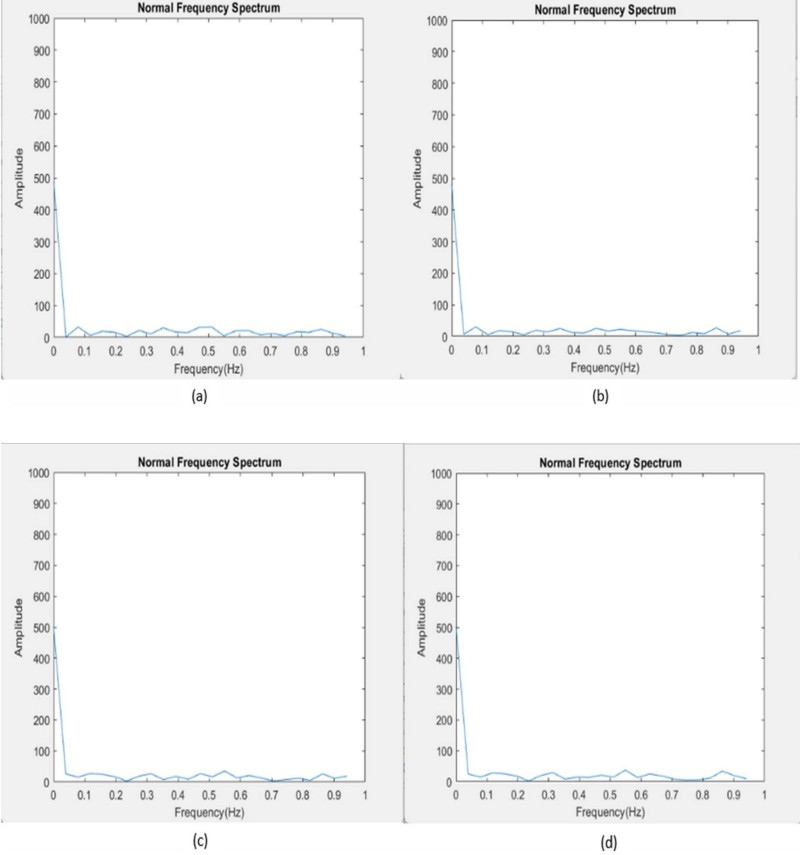

**Figure 14.** *Cont.*

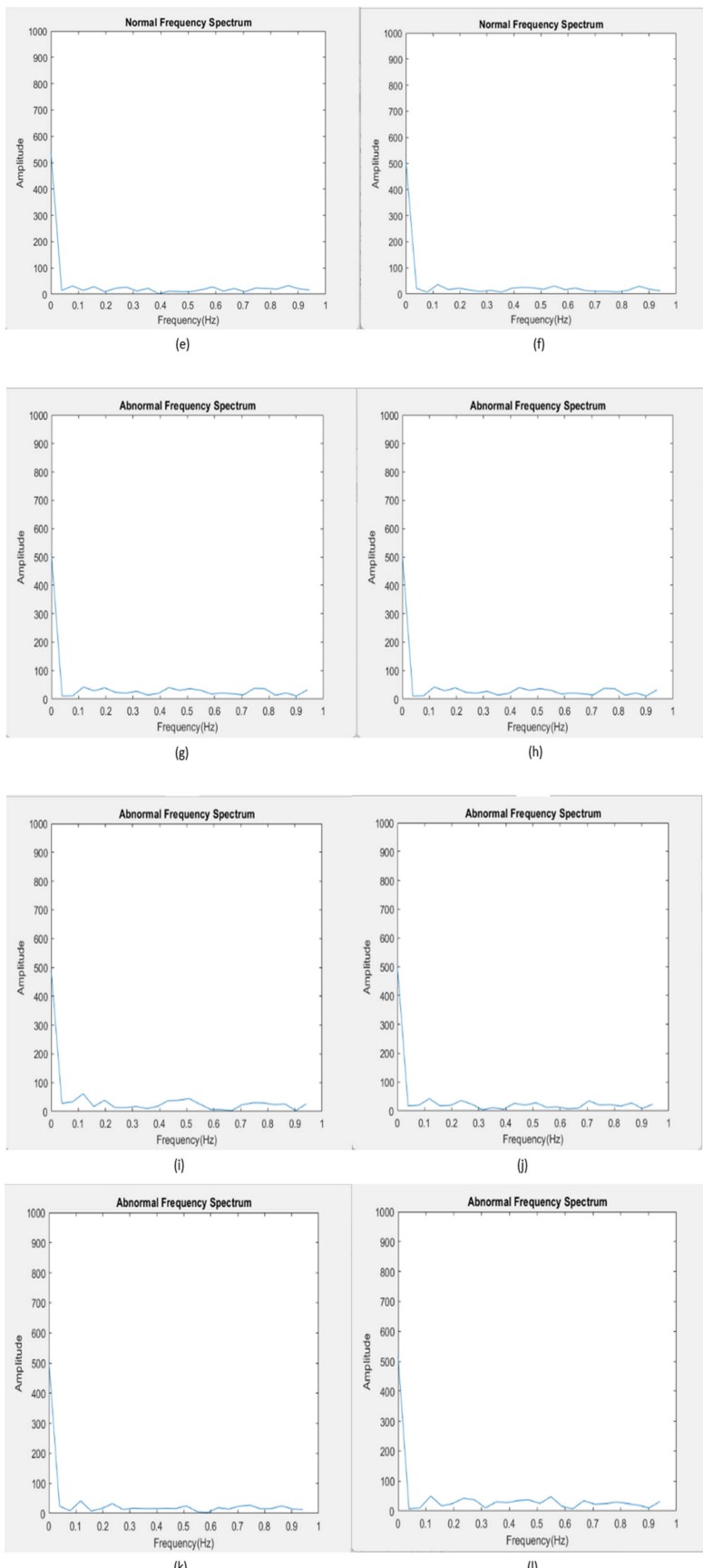

**Figure 14.** Segmented FFT spectrums for both normal and abnormal data sets: (**a–f**) segmented normal FFT spectrums, (**g–l**) segmented abnormal FFT spectrums.

It is clear from comparing the series of Figure 14 with Figures 6 and 7 that segmented FFT can only show a dominant low frequency. Other frequencies are not distinguishable from the spectrum, which confirms that FFT is not efficacious for trend detection use.

## 9. Conclusions

In this paper, several methods of fault trending detection have been trialled, including a control chart in SPC, FFT, and CWT. Vibration data sets were obtained from an automated trampoline webbing machine. The efficacy of these methods was compared by processing the experimental data sets through each of these methods. CWT has its advantage in the field of analysing signals in time and frequency domain. Fault conditions can be identified from changes in frequencies, and hence it is often used as the tool to capture a fault that has already occurred. However, frequency information alone does not imply the timing of a fault occurring, and hence is not sufficient to provide early warning of a fault occurring.

While CWT scalogram showed promising indicative outcomes, the CWT scalograms were difficult to use for the pulse counting process proposed. Instead of analysing the whole set of data points, the experimental data is divided into several time segments. In each time segment, synchrosqueezed scalograms are generated and the dominant frequency ranges are identified. The changing number of pulses in the dominant frequency ranges, especially the presence of troughs and turning points in the time segment graphs, is a good indicator of when a fault is about to happen in that time segment.

Based on the evaluation of efficacy, a novel method of analysing signals and predicting faults has been developed using the data-rich capability of Industry 4.0. The proposed digital signal processing method for trend prediction requires continuous performance data capture and analysis. Once the fault trend is recognised, preventive measures can be taken to stop or adjust the machine to eliminate the production of rejects and save time or re-work.

This research shows that there are a lot of numerical variation phenomena in the signal stream leading to a fault. To understand these variations by a computational process rather than by graph observation, machine deep learning, which has emerged as a powerful signal pattern recognition tool, is a reasonable direction for future research. The application of a deep learning algorithm often requires reliable input features to train the system before it can functionally predict or identify the unhealthy condition of the machine. The proposed algorithm has its merits for extracting fault information existing in transient signals, and thus can be integrated with deep learning algorithms in future studies.

**Author Contributions:** Composition, S.F.; revision, J.P.T.M. All authors have read and agreed to the published version of the manuscript.

**Funding:** This research received no external funding.

**Institutional Review Board Statement:** Not applicable.

**Informed Consent Statement:** Not applicable.

**Data Availability Statement:** The data presented in this study are available from the corresponding author upon request. The data are not publicly available due to the further construction of the weaving machine.

**Conflicts of Interest:** The authors declare no conflict of interest.

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
