# Peer review of "Efficacy Study of Fault Trending Algorithm to Prevent Fault Occurrence on Automatic Trampoline Webbing Machine"

_applsci, doi:10.3390/app12031708_

Round 1

Reviewer 1 Report

The paper tests several methods for fault trend detection, including control charts in SPC, FFT, and CWT. The effectiveness of the presented methods was tested by analyzing data obtained from an automated trampoline tape machine. A new method to analyze the signals and predict faults has been proposed. The experimental data were divided into several time segments. Synchrosqueezed scalograms are generated in each time segment, and the dominant frequency ranges are identified. The changing number of pulses in the dominant frequency fields, especially troughs and turning points in the time segment graphs, is treated as an indicator of a fault about to happen in that time segment.

General comments:

The work is interesting and touches on important practical issues but it is unclear how the simulated increase in resistance to rapier movement mimics the missing hook or the lack of thread caught on the hook. Therefore, it would be good to supplement the study with higher cases.

Detailed comments:

Axis descriptions are missing from some charts.

Some punctuation marks are missing: for example, a period at the end of a sentence before Conclusion.

Author Response

Dear reviewer,

Regards, 

Shi Feng John Mo 

Reviewer 2 Report

Dear Authors

You took care of the topic "Efficacy Study of Fault Trending Algorithm (??) to Prevent Fault Occurrence on Automatic Trampoline Webbing Machine ". At the outset, it should be emphasized that despite the high complexity of the studied object, you managed to achieve the intended goal. Nevertheless, according to the reviewer, you did not avoid a few defects, the consideration of which may have increased the quality of the manuscript. So they will be listed below one by one.

1) Manuscript title is too confusing because the word Fault occurs twice and in addition, is only one Algorithm tested for effectiveness?

2) The literature review is incomplete as not all items listed in REFERENCES are cited in the manuscript (e.g. Ding, Josso, Shore, check the others). In the cited position (Silva et al.), the time series of the casing acceleration data was compared using its Fourier spectrum and its synchrosqueezed CWT scalogram, which was not noticed by the Authors. Also, some of the best and newest publications dealing with this issue (synchrosqueezed CWT) are missing, such as:

Wang, L. Yang, X. Chen, et al .: Nonlinear squeezing time-frequency transform and application in rotor rub-impact fault diagnosis, J. Manuf. Sci. E.–T. ASME 139 (10) (2017) 101005, doi: 10.1115 / 1.4036993.

Tong, X. Chen, S. Wang: Nonlinear Squeezing Wavelet Transform for Rotor Rub-impact Fault Detection, in: Model Validation and Uncertainty Quantification, 3, Springer, 2019, pp. 21-29, doi: 10.1007 / 978-3-319-74793-4_4.

Yu, H. Ma, H. Han, et al .: Second order multi-synchrosqueezing transform for rub-impact detection of rotor systems, Mech. Mach. Theory 140 (2019) 321–349, doi: 10.1016 / j.mechmachtheory.2019.06.

It was also possible to indicate other very important literature items discussing this method in conjunction with AI methods (artificial intelligence, e.g. ANN), or with expert systems, such as:

Md Junayed Hasan, Akhand Rai, Zahoor Ahmad: A Fault Diagnosis Framework for Centrifugal Pumps by Scalogram-Based Imaging and Deep Learning. IEEE Access, vol. 9, pp. 58052-58066, 2021, doi: 10.1109 / ACCESS.2021.3072854.

Benkedjouh, Noureddine Zerhouni, S Rechak: Tool wear condition monitoring based on continuous wavelet transform and blind source sep. International Journal of Advanced Manufacturing Technology, 2018, 97 (7), pp. 1 - 3. ffhal-03053021f

Witulska, P. Stefaniak, B. Jachnik: Recognition of LHD Position and Maneuvers in Underground Mining Excavations — Identification and Parametrization of Turns. Appl. Sci. 2021, 11 (13), 6075; https://doi.org/10.3390/app11136075

Burriel-Valencia, R. Puche-Panadero, J. Martinez-Roman: Automatic Fault Diagnostic System for Induction Motors under Transient Regime Optimized with Expert Systems. Electronics 2019, 8 (1), 6; https://doi.org/10.3390/electronics8010006

Zhi'An Song; YuFeng Song: A Method of Gear Fault Diagnosis Based on CWT and ANN. 2009 International Conference on Business Intelligence and Financial Engineering. 24-26 July 2009 DOI: 10.1109 / bife15749.2009.

In this way, you could indicate in Conclusion the directions of further work on improving the effectiveness of the algorithm aimed at preventing fault occurence on Automatic Trampoline Webbing Machine (which is rather a necessity).

3) The algorithm proposed by the authors is interesting, but in the opinion of the reviewer, it does not indicate the methods of selecting the optimal wavelets type and, in a sense, the time processing domain (since the proposal is to be an improvement of the existing methods). It is known that in engineering application, there is a problem of how to select a suitable wavelet, despite the fact that wavelet in CWT can be selected flexibly. But more in: Li Lia, Liangsheng Qu, Xianghui Liao: Haar wavelet for machine fault diagnosis. Mechanical Systems and Signal Processing 21 (2007) 1773-1786. In addition it is known from the literature that in wavelet analysis, signal processing techniques (among the time, frequency or time-frequency domains) are most effective because they have a time-frequency domain. Nevertheless, the low frequency range is well resolved in the frequency domain but has poor resolution and the high frequency range has good temporal resolution but poor frequency resolution

If possible, please refer to the above comments (important or less important in the opinion of the reviewer). Best regards, Reviewer.

Author Response

(The authors gave the same response as above.)

Reviewer 3 Report

THis manuscript presents interesting results and can be accepted. 

Author Response

(The authors gave the same response as above.)

Reviewer 4 Report

In the paper, an interesting and challenging problem of fault diagnostics employing data from vibration sensor installed on a manufacturing line is considered. Relying on recorded signals the authors are trying to predict the occurrence of possible faults using various signal processing techniques. While, in general, the content of the paper is within the scope of Applied Sciences journal, in its current form, it could not be accepted for publishing it. The main reason is that the research methodology and obtained results are poorly and very unclearly presented.

The main issues are as follows:

  • The way how the obtained data is presented is, unfortunately, unsatisfactory. From figures 3 to 7 it absolutely not clear what are the signals collected from the sensor and were they collected for the state of normal functioning of the device or not. The authors must start from providing on the same plot two signals – the one measured for the “normal” case (“no fault observed”) and the second for the abnormal one. Moreover, all further signal processing results should be presented for both signals. Currently, it is even not clear, for what kind of data (normal or abnormal) results presented in Figures 3 to 13 are obtained.
  • The authors are incorrectly implementing FFT. It is well known that standard numerical procedure would provide periodic spectrum output. With this respect, most of the discussion related to figures 4 and 5 is incorrect.
  • From the second paragraph of Section 6, it is not clear at all what is the reason for such data manipulations? Moreover, there is no consistency between the spectrum of the signal and its scalogramm. From Figs 4 and 5 is clear that maximal values of spectrum occur close to 0 Hz while in Figure 6 they appear near 1 Hz
  • The main finding related to possibility of fault detection rely on the distribution of so called “frequency points” (beginning of subsection 7.3). The authors insist that the highest number of pulses occurs in two segments, namely, 0.3-0.4 Hz and 0.7-0.8 Hz. Unfortunately, thorough analysis of Figs. 8 and 12 does not allow to unequivocally agree with this statement. There are also a lot of “pulses” in segments 0.5-0.6 and 0.6-0.7 Hz (especially, in Fig. 12). These issues should be very carefully addressed.

Author Response

(The authors gave the same response as above.)

Round 2

Reviewer 1 Report

I find the authors' responses satisfying. 

It just seems that the description under the drawings is odd. The graphs show Y values as a function of X, suggesting using this convention.

I recommend the article for publication after minor revision.

Author Response

Dear Reviewer, 

Regards, 

Shi Feng John Mo 

Reviewer 4 Report

Unfortunately, the revised version of the manuscript is still possessing serious flaws. They are as follows:

1) The authors have not provided any examples of signals regarding the “normal” case (“no fault observed”) and the abnormal one as it was suggested in my first comment.

2) From the response to the third comment, it is still not clear why such data preparation is performed, i.e., why 300 point segments were selected, why 50 data points are duplicated between them, etc. Moreover, if such manipulations are performed, it is highly likely that even conventional FFT might be enough to compute "the number of pulses in different frequency ranges of each time segment" and no further application of CWT or  synchronsqueezed WT is necessary. With this respect, the author must provide FFT results not for the whole time segment as it is done in Figures 4 and 5, but for such reduced time segments.

3) Along with frequency range distribution for "abnormal" data sets, analogous results for regular performance must be provided as counterparts of Figures 8 and 12.

4) In the response to the 5th comment, the authors mention that "Yes, 0.5-0.6 and 0.6-0.7 Hz or any other frequencies can have a lot of pulses but they are not unique, they appear in any types of data during the experiment, so it is not the desired information we want". In the current version of the manuscript, this statement is not supported at all. Thus, all the results presented in subsection 7.4 as well as the conclusion could not be considered as being scientifically rigorous. 

5) When describing Figure 11, it is mentioned that "These time segments combined with the points online graph can be compared with the recorded fault approaching time with the recorded time when the fault is about to happen. One abnormality can be observed, i.e., every time when there is a decreasing trend before a time segment following with an increasing trend, the fault is about to happen in that time segment." All these statements are not supported with any examples and/or justifications.

6) The quality of Figure 2 should be improved. LSM 303DLHC accelerometer and Arduino Mega 2560 should be marked on it. In Figure 1, hooks, strainer, and other important parts should be clearly marked.

Author Response

(The authors gave the same response as above.)

Round 3

Reviewer 4 Report

The authors have introduced remarkable improvements to the paper. It could be accepted for publishing in Applied Sciences journal.